# The role of passion and affect in adolescents' basketball participation: A self-determination theory perspective

Jiangchuan Wang, Yujie Mao, Peng Tang *

Department of Physical Education, Hohai University, Nanjing, China

* 19990014@hhu.edu.cn

## Abstract

This study examined adolescent basketball participation through the lens of Self-Determination Theory (SDT), focusing on the roles of passion (harmonious and obsessive) and affect (positive and negative), with gender and age as moderators. We collected data from 400 Chinese adolescents from south of China using convenience sampling ($M_{age}$ = 15.20, $SD$ = 1.99). Participants completed the Passion Scale and Positive and Negative Affect Schedule Scale (PANAS). Basketball participation was operationalized as self-reported engagement in basketball over the past seven days. Hypothesized pathways were tested using structural equation modeling (SEM) in Mplus, which included test of measurement invariance and moderation. Results revealed that harmonious passion positively predicted basketball engagement (β = .280, 95% CI [.127,.432], $p$ = .003) though enhanced positive affect. Specifically, harmonious passion was positively associated with positive affect (β = .495, 95% CI [.356,.635], $p$ < .001), which, in turn, predicted basketball engagement (β = .253, 95% CI [.132,.373], $p$ = .001). In contrast, obsessive passion was linked to negative affect (β = .196, 95% CI [.083,.310], $p$ = .004) but showed no direct effect on basketball participation. While no significant gender differences were observed, age differences emerged. Specifically, the pathways from both harmonious and obsessive passion to basketball engagement were stronger among junior high school students compared to their senior high school counterparts. This study revealed that harmonious passion and positive affect play pivotal roles in sustaining adolescent basketball participation, whereas obsessive passion's negative emotional consequences may undermine engagement. The findings underscore the importance of interventions designed to promote harmonious passion and positive affect, particularly for senior high students facing academic pressures. Given the cross-sectional design and sampling from southern China, the generalizability of our findings to other cultural contexts or regions of China should be interpreted with caution. Additionally, due to the less-than-ideal model fit in multi-group analyses examining gender as a moderator, results concerning gender differences require the same cautious approach.

**Data availability statement:** All data necessary to replicate the findings of this study are publicly available in the ScienceDB repository (https://doi.org/10.57760/sciencedb.28416) for academic research purposes only. Commercial use is strictly prohibited.

**Funding:** The author(s) received no specific funding for this work.

**Competing interests:** NO authors have competing interests.

## Introduction

Basketball is widely regarded as one of the most beloved sports because of its dynamic nature, competitive spirit, and worldwide popularity [1]. This sport not only entertains but also offers substantial health benefits to participants [2,3]. Prior research consistently highlights basketball's positive effects on physical and mental health, such as improving fitness, muscular strength, and cognitive performance [4]. Additionally, basketball fosters social skills through teamwork and communication [5].

Sports, such as basketball, play an important role in China's youth development programs [6]. The Chinese government actively promotes basketball in schools across the country, such as setting up basketball in physical education classes and after-school activities [5]. In addition, local communities organize youth basketball competitions every year, and sports centers provide free basketball coaching for teenagers. These programs help adolescents stay healthy and develop teamwork skills [7]. Yet, it remains unclear how greater basketball exposure in school curricula influences adolescents' engagement in this sport.

Previous Chinese research has primarily focused on competitive basketball and basketball-related injuries, with relatively few studies investigating the influencing factors and underlying mechanisms of adolescent basketball participation [8–10]. This study examines the factors influencing basketball participation among Chinese adolescents, with a specific focus on the role of passion (harmonious passion and obsessive passion) and affect (positive affect and negative affect) based on Self-Determination Theory (SDT). Additionally, the study investigates potential gender- and age-related differences in these relationships by developing stratified models.

## Literature review

As basketball positively contributes to adolescents' physical and mental development [5], it has been widely incorporated into school curricula and extracurricular sports programs across China. Several factors influence basketball participation among adolescents, including individual motivation, emotional experiences, and sociocultural contexts. Research indicates that intrinsic motivation and positive affect play crucial roles in sustaining engagement, while external pressures and negative affect may hinder participation [11,12]. Additionally, gender and age differences further shape these dynamics, with boys and younger students typically exhibiting higher involvement due to sociocultural norms and academic demands [13,14]. The following sections explore the roles of motivation and affect in basketball participation, followed by an analysis of gender- and age-related disparities.

### The dualistic model of passion

Self-Determination Theory (SDT) establishes that high-quality motivation drives sustained participation in physical activities when three basic psychological needs are met: autonomy, competence, and relatedness. According to SDT, motivation drives people to participate in physical activity [15]. SDT explains different types of motivation. Intrinsic motivation means playing basketball because it is fun, whereas external

motivation means playing for rewards or outcomes [11,12]. SDT is widely used in the field of sport, and intrinsic motivation has a stronger positive effect on sports participation than external motivation [16]. In basketball, motivation matters for both athletes and students.

SDT suggests that meeting basic psychological needs is key to maintaining participation in activities [11]. Building on SDT, passion is a strong attraction to certain activities [17]. According to the Dualistic Model of Passion (DMP), people with passion willingly spend time and energy on what they love [18], and passion connects closely with motivation and helps people start and continue activities [19].

The DMP identifies two types of passion: harmonious and obsessive passion [17]. Obsessive passion means feeling forced to do an activity, which can cause problems. Harmonious passion means choosing to do something freely without pressure [18]. Research shows that harmonious passion leads to better sports participation; people with this passion practice more and enjoy activities more [20]. They also feel more focused during activities and more satisfied afterward [21].

While prior studies have examined passion in sports [20], basketball presents unique motivational dynamics due to its team-based nature and high social visibility. Recent evidence suggests that in competitive basketball, obsessively-passionate players reported more aggressive behavior [22]. However, few studies have explored how these passion types operate among adolescent recreational players, particularly in collectivist cultures like China, where academic pressures may alter passion trajectories. This gap warrants investigation because basketball participation in schools often serves dual purposes-skill development and stress relief-which may differentially predict harmonious versus obsessive passion.

This theoretical integration proves particularly relevant for understanding adolescent sports participation in collectivist cultures like China. The compatibility between harmonious passion and SDT's autonomous motivation suggests that basketball programs fostering player autonomy and personal meaning may be most effective for sustaining engagement. Meanwhile, the tension between obsessive passion and need satisfaction highlights potential risks of overemphasizing performance outcomes or external rewards in youth sports development.

Specifically, the connection focuses on SDT's three basic psychological needs: autonomy, competence, and relatedness. Harmonious passion has a voluntary and balanced nature that directly satisfies these three needs [17]. When adolescents play basketball out of harmonious passion, they feel autonomous and gain a sense of achievement from improving their skills. Satisfying these basic needs triggers positive affect, such as joy and excitement. This affective mechanism is what drives sustained basketball participation.

In contrast, obsessive passion conflicts with these basic needs [19]. Adolescents with obsessive passion play basketball to avoid guilt or gain approval. As a result, they lack a sense of autonomy and often feel anxious about failure, which weakens their sense of competence. This harms their sense of relatedness, which explains why obsessive passion links to more negative affect.

## Passion, affect, and basketball participation

Affects are psychological states that arise in response to experiences and influence behaviors [23]. In sports, affect plays a critical role in an individual's performance, engagement, and long-term participation [24]. However, previous studies mainly explored the impact of sports activities on affect [25,26]. Relatively little research in sports psychology has examined how affective factors like positive and negative affect influence individuals' behaviors, decision-making, and persistence in physical activities [27]. Positive affect, such as excitement and pride, is linked to greater effort and sustained participation, while negative affect, such as stress and disappointment, may reduce motivation and lead to withdrawal.

Based on the integrated framework of SDT and DMP, the two types of passion show mechanistic differences in their effects on affective states, and these differences rely on the mediating role of basic psychological need satisfaction (a core concept of SDT). Researchers have confirmed that this mediating path explains why passion types correlate with distinct affective outcomes in sport contexts [17,19].

                                                          

Research based on DMP distinguishes two types of passion with different effects on sports engagement [20]. Harmonious passion, characterized by voluntary and balanced involvement, is associated with long-term participation and positive affect experiences during play [19]. In the basketball context, this passion comes from an individual's voluntary choice of basketball activities, not external pressure. Thus, this autonomy promotes the satisfaction of SDT's core psychological needs (autonomy, competence, and relatedness) during basketball participation [28,29]. Studies have shown that satisfied psychological needs directly trigger positive affect (e.g., pleasure, a sense of accomplishment) and suppress the generation of negative affect [19,30].

In contrast, obsessive passion, marked by internal pressure to engage, often acts though the "psychological conflict-need deprivation-negative affect" and correlates with affect conflict and higher dropout risk [19]. Although obsessive passion shows high investment in basketball, it essentially belongs to controlled motivation, which conflicts with the sense of autonomy [17]. Even if obsessive passion occasionally triggers positive affect, it is more likely to induce negative affect in sports contexts.

Affect responses appear to mediate how passion influences sport participation. Studies find that positive affect, such as enjoyment and satisfaction, enhances skill development and reinforces commitment to the sport [27]. Negative affect, such as anxiety and frustration, can undermine performance motivation and lead to disengagement [31]. These patterns are particularly evident; harmonious passion tends to predict sustained involvement through positive affect experiences, while obsessive passion may lead to affect exhaustion and reduced participation over time among adolescents.

The combined evidence suggests that passion shapes basketball participation both directly and through its affective consequences. Understanding these relationships can inform strategies to promote healthy and lasting engagement in youth basketball.

## Cultural context of Chinese adolescent basketball participation

The basketball participation patterns of Chinese adolescents emerge from a complex interplay of sociocultural factors that remain understudied in global sports literature [32]. Compared to many Western education systems that structurally integrate sports into school curricula, China's distinctive academic pressures, gender expectations, and resource distribution patterns present different challenges for youth basketball participation [33].

The Chinese education system's intense focus on the Gaokao college entrance examination creates significant barriers to sustained sports participation. Research indicates a dramatic decline in physical activity levels during the senior high school years [34]. This academic pressure particularly affects basketball participation due to the sport's demanding time requirements for team coordination and skill development [35]. The "Double Reduction" policy, while aiming to alleviate student burdens, has shown limited effectiveness in reversing this trend for organized team sports.

## Gender and age differences in basketball

However, significant gender disparities persist in youth physical activity, especially in football and basketball, with boys demonstrating consistently higher engagement levels than girls [36]. The imbalance primarily stems from sociocultural stereotypes that position basketball as a masculine activity [37], resulting in unequal access to training resources and fewer competitive opportunities for female students. Current interventions show promising yet limited success in addressing this gap. While progressive schools have implemented gender-sensitive coaching methods and established girls' basketball leagues, deeply rooted cultural perceptions continue to hinder equal participation [38].

Age difference in physical activity becomes particularly evident when comparing different age students [39,40]. For example, Chinese senior high students participate relatively less due to the intense academic pressure from college entrance exams. Schools and parents generally prioritize academic studies, leading to reduced time for physical activities and fewer opportunities for basketball participation during this critical stage [41]. Therefore, this is a pivotal point where many adolescents, regardless of initial interest or talent in basketball, are effectively forced to reduce sustained participation in this sport.

Prior researches highlight the complex interplay of factors influencing adolescent basketball participation [42]. Gender disparities are rooted in sociocultural perceptions of basketball as a male-dominated sport [37], with boys receiving more encouragement and opportunities than girls. Furthermore, the research identifies a pronounced age-related decline in participation during senior high school years, when academic pressures from college entrance exams substantially reduce students' available time for sports [43].

The combination of these gender and age factors creates distinct participation patterns that vary across developmental stages. These insights contribute to our understanding of the multidimensional barriers to youth basketball engagement and underscore the need for targeted interventions that address both sociocultural norms and structural constraints in China's educational environment.

### The current research

This study examines three key research questions: (a) how passion influences adolescents' basketball participation, (b) the mediating role of affect in this relationship, and (c) potential gender differences in these associations. Based on theoretical foundations, we hypothesize that (as shown in Fig 1):

*H1a*: Harmonious passion positively influences adolescents' positive affect.
*H1b*: Harmonious passion negatively influences adolescents' negative affect.
*H1c*: Obsessive passion negatively influences adolescents' positive affect.
*H1d*: Obsessive passion positively influences adolescents' negative affect.
*H2a*: Harmonious passion positively influences adolescents' basketball engagement.
*H2b*: Obsessive passion negatively influences adolescents' basketball engagement.
*H3a*: Positive affect positively influences adolescents' basketball engagement.
*H3b*: Negative affect negatively influences adolescents' basketball engagement.

## Methods

### Participants

This research recruited a total of 450 participants using convenience sampling methodology. The blank questionnaire or the questionnaire with same response [44] would be excluded so the final sample is 400 ($M = 15.200$, $SD = 1.990$, Male = 112, Female = 286, Unknown = 2). The sample size of this study satisfies the minimum requirement for structural equation modeling as suggested by prior research [45].

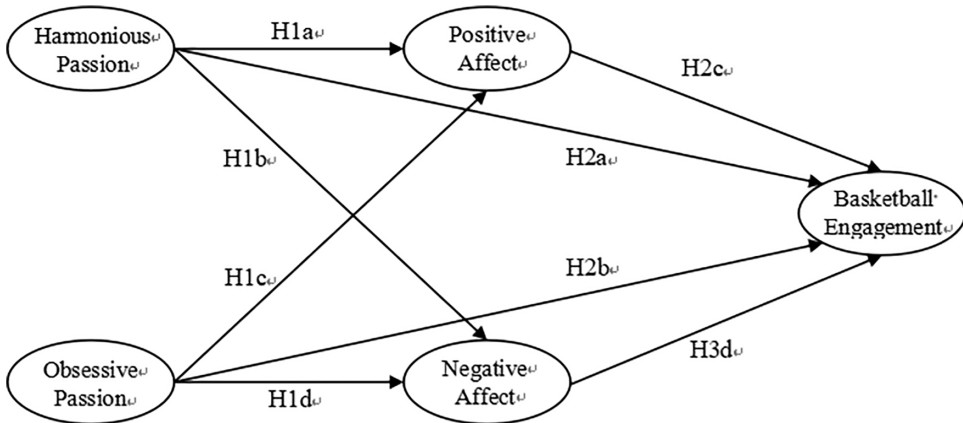

**Fig 1. Hypothesis model.**

## Measures

**Passion.** This study employed the Chinese version of the Passion Scale [46], which contains 17 items measuring two dimensions: harmonious passion (6 items, α = .860) and obsessive passion (6 items, α = .820). Participants rated each item on a 7-point Likert scale ranging from 1 ("Not agree at all") to 7 ("Very strongly agree"). We adapted the original scale by replacing "academic activities" with "basketball" to suit our research context. The modified scale demonstrated excellent reliability, with harmonious passion showing α = .908, while obsessive passion yielded α = .838. Additionally, we conducted confirmatory factor analysis (CFA) to test the scale's validity. The CFA results indicate structural validity of the modified scale are acceptable ($\chi^2$ = 223.269, CFI = .918, TLI = .892, RMSEA = .096, SRMR = .066).

**Affect.** This study measured positive and negative affects using the Positive and Negative Affect Schedule (PANAS), a well-validated scale developed by Watson, Clark, and Tellegen [47]. The PANAS consists of two 9-item subscales assessing Positive Affect (PA) and Negative Affect (NA) in Chinese version. Participants rated the extent to which they experienced each affect (e.g., "excited", "upset") on a 5-point Likert scale (1 = very slightly or not at all, 5 = extremely). The PANAS demonstrates strong reliability (Cronbach's α = .84 − .90). Internal consistency (PA: Cronbach's α = .852, NA: Cronbach's α = .864) typically exceeds.85 for both subscales in this study. The CFA results showed acceptable structural validity: $\chi^2$ = 417.788, CFI = .915, TLI = .902, RMSEA = .074, and SRMR = 0.068, all meeting standard fit criteria. The scale has been widely used in adolescent and sports research [24,48].

**Basketball participation.** Participants self-reported their basketball participation during the previous seven days by answering two specific questions: (1) "Besides school-arranged basketball activities, how many days did you participate in basketball activities in the past 7 days?" and (2) "Besides school-arranged basketball activities, how long did you participate in basketball activities each time in the past 7 days?" A composite score of basketball engagement was calculated as the product of the days and duration (Days × Duration). This score was standardized using z-scores and extreme values beyond ±3.29 were excluded [49]. In SEM, we modeled this composite as a latent variable with the two items as indicators, following a confirmatory factor analysis approach to examine their association with the underlying construct. The modified scale demonstrated excellent reliability (α = .930). To support criterion validity, basketball participation showed a significant positive correlation with harmonious passion ($r$ = .272, $p$ < .001), consistent with theoretical expectations.

**Demographic variables.** Demographic variables included participants' school, gender and age.

## Procedure

This study was approved by the university's ethics committee (IRB: HHU-2025-01) and collected the data from a junior high school and a senior high school in south of China between April and May, 2025. With the acquisition of the school and parents, this research recruited a total of 450 participants using the paper version of the questionnaire. Parents provided written informed consent for their children's participation in the research questionnaire. Participants completed the questionnaires in the classroom under teachers' guidance, and the process took 10–15 minutes. To increase the response rate, participants received a small gift for completing the survey. All data necessary to replicate the findings of this study are publicly available in the ScienceDB repository (https://doi.org/10.57760/sciencedb.28416) for academic research purposes only.

## Statistical analysis

Data analysis was conducted using SPSS 25.0 for descriptive statistics, correlation analysis, internal consistency assessment (Cronbach's α), and multivariate analysis of variance (MANOVA). For scale validation, Confirmatory Factor Analysis (CFA) and hypothesis testing of the structural model were performed in Mplus 8.3. Missing data were handled using Mplus's default Full Information Maximum Likelihood method (FIML).

Model fit was evaluated using multiple indices: chi-square statistic ($\chi^2$), Comparative Fit Index (CFI), Tucker-Lewis Index (TLI), Root Mean Square Error of Approximation (RMSEA), and Standardized Root Mean Square Residual (SRMR). Good model fit was indicated by CFI and TLI values $\geq 0.95$, RMSEA $\leq 0.06$, and SRMR $\leq 0.08$. These thresholds ensured robust validation of the measurement and structural models [50].

## Results

### Multi-Collinearity and common methods bias

Multicollinearity occurs when predictor variables are highly correlated, potentially obscuring their true relationships with the outcome variable [51]. This study evaluated multicollinearity using variance inflation factors (VIF). A VIF exceeding 10 indicates severe multicollinearity requiring remediation [52]. Our results showed VIF values ranging from 1.022 to 2.001 (Table 1), confirming acceptable levels of multicollinearity among predictors.

This study assessed common method bias using Harman's single-factor test. The results showed that the cumulative percentage of variance explained by the first factor was 23.191%, which is below the critical threshold of 50%. This indicates that common method bias was not a serious concern in our data [53].

### Descriptive statistics

Descriptive statistic results are shown in Table 2, based on the result of the K-S test, all variables are non-normal distributed ($p = .001-.033$) except for positive affect ($p = .077$).

Harmonious passion positively correlates with obsessive passion ($r = .486$, $p < .001$), positive affect ($r = .419$, $p < .001$), and basketball engagement ($r = .272$, $p < .001$), and basketball participation ($r = .531$, $p < .001$). However, harmonious passion negatively correlates to gender ($r = -.281$, $p < .001$), age ($r = .294$, $p < .001$), and not correlates to negative affect ($r = -.021$, $p = .682$).

Obsessive passion positively correlates with positive affect ($r = .193$, $p < .001$) and basketball engagement ($r = .164$, $p = .001$). However, obsessive passion negatively correlates to gender ($r = -.198$, $p < .001$), and not correlates to negative affect ($r = -.098$, $p = .056$) and age ($r = .051$, $p = .313$).

Positive affect positively correlates with basketball engagement ($r = .225$, $p < .001$). However, positive affect negatively correlates to gender ($r = -.222$, $p < .001$), age ($r = -.321$, $p < .001$), and not correlates to negative affect ($r = .013$, $p = .540$).

Negative affect positively correlates with age ($r = .172$, $p < .001$). However, positive affect not correlates to basketball engagement ($r = -.031$, $p = .540$) and gender ($r = .099$, $p = .077$).

Basketball engagement not correlates to gender ($r = -.455$, $p < .001$) and age ($r = -.312$, $p < .001$).

### Gender and age differences in the factors

We used MANOVA in SPSS to examine gender and age differences in study variables (junior vs. senior high school). Table 3 shows significant gender differences for all factors except negative affect. Age differences emerged across all variables between junior and senior high students.

**Table 1. Results of VIF.**

|   | Variables | Tolerance | VIF |
|---|---|---|---|
| 1 | Harmonious Passion | 0.979 | 1.022 |
| 2 | Obsessive Passion | 0.962 | 1.039 |
| 3 | Positive Affect | 0.500 | 2.001 |
| 4 | Negative Affect | 0.505 | 1.980 |

**Table 2. Results of descriptive statistics.**

| | Variables | M | SD | K-S Test | 1 | 2 | 3 | 4 | 5 | 6 |
|---|---|---|---|---|---|---|---|---|---|---|
| 1 | Harmonious Passion | 4.433 | 1.359 | .033 | | | | | | |
| 2 | Obsessive Passion | 2.414 | 1.067 | <.001 | .486*** | | | | | |
| 3 | Positive Affect | 2.976 | 0.773 | .077 | .419*** | .193*** | | | | |
| 4 | Negative Affect | 2.051 | 0.774 | <.001 | −.021 | .098 | .013 | | | |
| 5 | Basketball Engagement | −0.062 | 0.414 | <.001 | .272*** | .164** | .225*** | −.031 | | |
| 6 | Gender | / | / | / | −.281*** | −.198*** | −.222*** | .090 | −.455*** | |
| 7 | Age | 15.200 | 1.990 | / | −.294*** | .051 | −.321*** | .172** | −.312*** | .269*** |

Notes. * $p < 0.05$, ** $p < 0.01$, *** $p < 0.001$

**Table 3. Results of MANOVA.**

| Variables | Male M (SD) | Female M (SD) | F | p | Junior high school student M (SD) | Senior high school student M (SD) | F | p |
|---|---|---|---|---|---|---|---|---|
| Harmonious Passion | 5.056 (1.491) | 4.171 (1.218) | 18.961 | <.001 | 4.948 (1.330) | 4.081 (1.266) | 29.998 | <.001 |
| Obsessive Passion | 2.742 (1.190) | 2.244 (0.979) | 20.343 | <.001 | 2.332 (0.967) | 2.416 (1.123) | 4.091 | .044 |
| Positive Affect | 3.240 (0.808) | 2.870 (0.736) | 7.304 | .007 | 3.361 (0.715) | 2.790 (0.755) | 29.462 | <.001 |
| Negative Affect | 1.954 (0.817) | 2.098 (0.768) | 0.574 | .449 | 1.901 (0.699) | 2.158 (0.819) | 6.450 | .012 |
| Basketball Engagement | 0.227 (0.678) | −0.174 (0.115) | 63.692 | <.001 | 0.103 (0.574) | −0.167 (0.200) | 40.197 | <.001 |

## Hypothesis testing

We tested our hypotheses using structural equation modeling (SEM). The model showed good fit with the data: $\chi^2 = 993.442$, CFI = .904, TLI = .894, RMSEA = .057, and SRMR = .063. All fit indices met recommended standards, indicating the model adequately represented our data [54]. Results as illustrated in Fig 2, our findings support the hypothesized relationships.

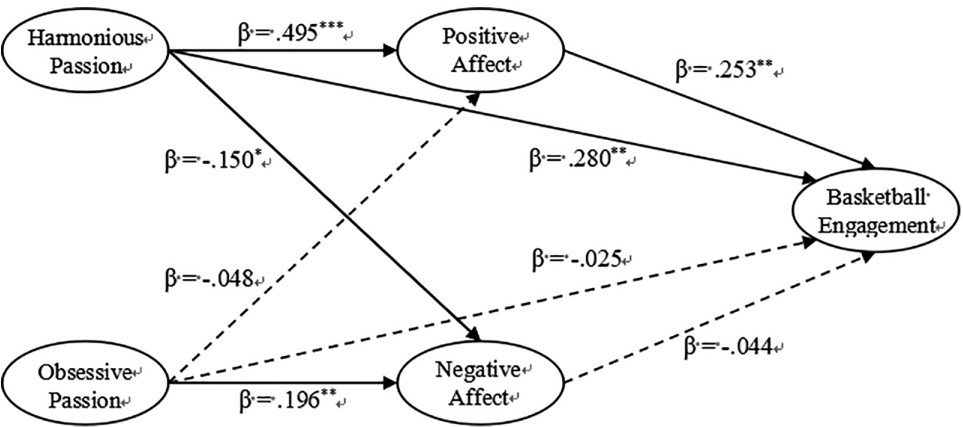

**Fig 2. Structural equation modeling results.** *Note.* Solid line indicates path is significant. Dashed lines indicate path is not significant.

H1a-b: harmonious passion has a positively significant impact on positive affect (β = .495, 95% CI [.356,.635], $p$ < .001) and negatively significant impact on negative affect (β = −.150, 95% CI [−.250, −.051], $p$ = .013).

H1c-d: obsessive passion has no significant impact on positive affect (β = −.048, 95% CI [−.180,.084], $p$ = .549) and negatively significant impact on negative affect (β = .196, 95% CI [.083,.310], $p$ = .004).

H2a-b: harmonious passion has a positively significant impact on basketball engagement (β = .280, 95% CI [.127,.432], $p$ = .003), whereas obsessive passion has no significant impact on basketball engagement (β = −.025, 95% CI [−.167,.118], $p$ = .774).

H3a-b: positive affect has a positively significant impact on basketball engagement (β = .253, 95% CI [.132,.373], $p$ = .001), whereas negative affect has no significant impact on basketball engagement (β = −.044, 95% CI [−.138,.051], $p$ = .446).

### The moderation role of gender and age group

We investigated gender and age (junior vs. senior high school) as potential moderators in the SDT framework, focusing on relationships between passion types (harmonious/obsessive) and affective states (positive/negative affect). After confirming measurement invariance across groups (ΔCFI < .01, RMSEA < .08, SRMR < .08), we examined moderation effects on six core pathways: (1) harmonious passion to positive affect, (2) harmonious passion to negative affect, (3) obsessive passion to positive affect, (4) obsessive passion to negative affect, and (5) positive affect to basketball engagement, (6) negative affect to basketball engagement. The invariance tests followed prior criteria [55], with all models demonstrating acceptable configural, metric, and scalar invariance (Table 4).

We performed separate SEM analyses to examine gender and educational stage moderation effects. For gender moderation, the model showed marginal fit ($\chi^2$ = 1842.300, CFI = .845, TLI = .840, RMSEA = .071, SRMR = .079). For educational stage moderation, similar results emerged ($\chi^2$ = 1819.088, CFI = .846, TLI = .841, RMSEA = .070, SRMR = .084). While CFI/TLI values were slightly below the 0.900 threshold, RMSEA values met acceptable standards.

Table 5 shows significant moderation effects of age on the relationship between harmonious passion and basketball engagement, obsessive passion and basketball engagement. Specially, for male participants, both harmonious passion (β = .264, $p$ = .004) and obsessive passion (β = 248, $p$ = .009) significantly predicted basketball engagement, while neither pathway reached significance for female participants (HP: β = −.041, $p$ = .720. OP: β = .062, $p$ = .596).

## Discussion

This study employed SDT and DMP to examine factors (passion and affect) influencing adolescent basketball participation, with particular attention to gender and age as moderating variables. The findings demonstrated that harmonious passion positively predicted basketball engagement through enhanced positive affect, whereas obsessive passion primarily operated through increased negative affect without demonstrating a direct effect on participation.

### The role of harmonious passion and positive affect in basketball participation

The study found that harmonious passion directly and positively predicts basketball participation among Chinese adolescents. This aligns with SDT's emphasis on autonomous motivation, where harmonious passion reflects a voluntary and integrated engagement with basketball [56,57]. In China's context, where sports are often promoted for holistic development (e.g., school curricula and government initiatives), harmonious passion enables adolescents to view basketball as a personally meaningful activity rather than an obligation. For example, students who play basketball out of genuine interest (harmonious passion) are more likely to sustain participation despite academic pressures, as the activity aligns with their identity and values. Schools and coaches should prioritize fostering intrinsic motivation by creating autonomy-supportive environments [58]. For instance, offering student-led basketball clubs or choice-based training drills could enhance harmonious passion.

Harmonious passion also indirectly boosts participation by enhancing positive emotions, which in turn predict higher engagement. In China, where academic stress is pervasive, basketball's role as a stress-reliever is critical. Adolescents

**Table 4. Measurement invariance of the factors.**

| Gender | | | | | | | | | Age | | | | | | | | |
|---|---|---|---|---|---|---|---|---|---|---|---|---|---|---|---|---|---|
| Model | | $\chi^2$ | df | RMSEA | CFI | SRMR | △CFI | △RMAEA | Model | | $\chi^2$ | df | RMSEA | CFI | SRMR | △CFI | △RMAEA |
| 1 | Configural Invariance | 184.009 | 18 | .215 | .884 | .063 | / | / | 5 | Configural Invariance | 191.254 | 18 | .219 | .881 | .061 | / | / |
| 1 | Metric Invariance | 186.559 | 23 | .189 | .886 | .067 | .002 | −.016 | 5 | Metric Invariance | 195.771 | 23 | .194 | .881 | .070 | <.001 | −.025 |
| 1 | Scalar Invariance | 193.227 | 28 | .172 | .885 | .071 | .001 | −.017 | 5 | Scalar Invariance | 206.040 | 28 | .178 | .877 | .075 | −.004 | −.016 |
| 2 | Configural Invariance | 96.908 | 18 | .148 | .915 | .051 | / | / | 6 | Configural Invariance | 92.008 | 18 | .143 | .924 | .052 | / | / |
| 2 | Metric Invariance | 106.659 | 23 | .135 | .910 | .064 | −.005 | −.013 | 6 | Metric Invariance | 98.564 | 23 | .128 | .922 | .060 | −.002 | −.015 |
| 2 | Scalar Invariance | 137.312 | 28 | .140 | .882 | .090 | .028 | .005 | 6 | Scalar Invariance | 130.350 | 28 | .135 | .895 | .083 | .007 | −.027 |
| 3 | Configural Invariance | 195.217 | 54 | .115 | .909 | .055 | / | / | 7 | Configural Invariance | 191.110 | 54 | .113 | .908 | .058 | / | / |
| 3 | Metric Invariance | 209.62 | 62 | .109 | .905 | .077 | −.006 | −.004 | 7 | Metric Invariance | 203.722 | 62 | .107 | .905 | .075 | −.006 | −.003 |
| 3 | Scalar Invariance | 220.296 | 70 | .104 | .903 | .087 | −.005 | .010 | 7 | Scalar Invariance | 217.900 | 70 | .103 | .901 | .076 | −.004 | .001 |
| 4 | Configural Invariance | 150.367 | 54 | .095 | .944 | .042 | / | / | 8 | Configural Invariance | 194.096 | 54 | .114 | .919 | .047 | / | / |
| 4 | Metric Invariance | 166.968 | 62 | .092 | .939 | .059 | −.005 | −.003 | 8 | Metric Invariance | 229.329 | 62 | .116 | .903 | .081 | .002 | −.016 |
| 4 | Scalar Invariance | 180.600 | 70 | .089 | .936 | .058 | −.003 | −.003 | 8 | Scalar Invariance | 254.557 | 70 | .115 | .893 | .084 | −.001 | −.010 |

*Note.* Model 1 demonstrates the measurement invariance of harmonious passion across gender groups.

Model 2 demonstrates the measurement invariance of obsessive passion across gender groups.

Model 3 demonstrates the measurement invariance of positive affect across gender groups.

Model 4 demonstrates the measurement invariance of negative affect across gender groups.

Model 5 demonstrates the measurement invariance of harmonious passion across age groups.

Model 6 demonstrates the measurement invariance of obsessive passion across age groups.

Model 7 demonstrates the measurement invariance of positive affect across age groups.

Model 8 demonstrates the measurement invariance of negative affect across age groups.

with harmonious passion experience joy and excitement during play, reinforcing their commitment. This is particularly relevant given China's "double reduction" policy, which aims to reduce academic burden and promote extracurricular activities [59,60]. Positive affect transforms basketball into a rewarding escape, counterbalancing stress. Coaches should design sessions that maximize enjoyment (e.g., small-sided games, peer collaboration) and integrate mindfulness techniques to amplify positive emotions linked to harmonious passion.

## The role of obsessive passion and negative affect in basketball participation

Contrary to expectations, obsessive passion showed no direct effect on basketball engagement. This suggests that in China's collectivist culture, where external pressures (e.g., parental expectations or peer comparisons) often drive obsessive passion, such motivation fails to sustain long-term participation. Specifically, students may develop obsessive passion due to academic pressure (such as physical education exams) or being forced to practice basketball in PE classes under the

**Table 5. The associations between factors and intention by gender and age.**

| Route | Male | | | Female | | | Comparison | | |
|---|---|---|---|---|---|---|---|---|---|
| | β | SE | p | β | SE | p | β | SE | p |
| Harmonious Passion→Positive Affect | .558 | .060 | <.001 | .347 | .097 | <.001 | .079 | .061 | .192 |
| Obsessive Passion→Positive Affect | .002 | .073 | .980 | −.016 | .089 | .861 | .010 | .063 | .876 |
| Harmonious Passion→Negative Affect | −.150 | .072 | .038 | −.109 | .068 | .107 | −.014 | .049 | .767 |
| Obsessive Passion→Negative Affect | .228 | .100 | .023 | .188 | .069 | .007 | .010 | .067 | .884 |
| Harmonious Passion→Basketball Engagement | .279 | .131 | .033 | .201 | .110 | .067 | .175 | .143 | .220 |
| Obsessive Passion→Basketball Engagement | −.055 | .130 | .672 | −.171 | .102 | .094 | .032 | .149 | .832 |
| Positive Affect→Basketball Engagement | .181 | .122 | .138 | .278 | .077 | <.001 | .126 | .268 | .638 |
| Negative Affect→Basketball Engagement | −.051 | .101 | .616 | .064 | .090 | .478 | −.160 | .221 | .470 |
| Route | Junior high school student | | | Senior high school student | | | Comparison | | |
| | β | SE | p | β | SE | p | β | SE | p |
| Harmonious Passion→Positive Affect | .344 | .083 | <.001 | .387 | .111 | <.001 | −.036 | .070 | .605 |
| Obsessive Passion→Positive Affect | .029 | .094 | .753 | .041 | .097 | .673 | −.004 | .072 | .959 |
| Harmonious Passion→Negative Affect | −.092 | .073 | .205 | −.051 | .102 | .621 | −.014 | .064 | .827 |
| Obsessive Passion→Negative Affect | .092 | .092 | .315 | .165 | .099 | .096 | −.039 | .077 | .612 |
| Harmonious Passion→Basketball Engagement | .264 | .092 | .004 | −.041 | .114 | .720 | .322 | .128 | .012 |
| Obsessive Passion→Basketball Engagement | .248 | .096 | .009 | .062 | .117 | .596 | .317 | .157 | .043 |
| Positive Affect→Basketball Engagement | .228 | .088 | .010 | .142 | .064 | .026 | .408 | .229 | .074 |
| Negative Affect→Basketball Engagement | .072 | .097 | .457 | −.041 | .065 | .524 | .223 | .254 | .380 |

requirement of schools or parents. Previous studies have pointed out that this kind of passion is associated with negative psychological factors and may lead to negative emotions [19]. For example, students forced to excel in basketball for scholarships or social status may disengage once rewards disappear or academic demands escalate. Programs should identify and mitigate external pressures. In addition, coaches and parents could shift focus from outcomes to personal growth and skill mastery to reduce obsessive passion's negative aspects [61].

Obsessive passion predicted higher negative affect, likely due to internal conflicts (e.g., guilt over neglecting studies or fear of failure). However, this emotional toll did not translate to reduced participation, possibly because Chinese adolescents may persist due to filial piety or "face" culture, even when emotionally drained. Schools should provide counseling to help students balance OP-driven activities with well-being. For instance, workshops on emotional regulation could mitigate obsessive passion's harmful effects.

### Gender and age difference in basketball participation

Contrary to initial expectations, the results revealed no significant gender differences in the relationships between passion types (harmonious and obsessive) and basketball engagement. This suggests that the motivational pathways through which passion influences sports participation may operate similarly for both male and female adolescents in the studied sample. The absence of gender differences challenges prevailing sociocultural stereotypes that position basketball as a male-dominated activity [37] and implies that, when provided with equitable opportunities, female adolescents may derive comparable motivational benefits from basketball participation.

Adolescents' behavior associates with environment and policy [62]. The lack of gender differences could be attributed to the increasing efforts in China to promote gender equality in sports, such as the implementation of gender-sensitive coaching and the establishment of girls' basketball leagues. These initiatives may have mitigated traditional barriers,

allowing female students to experience similar motivational and affective outcomes as their male counterparts. Additionally, the study's focus on school-based participation, where structured environments may reduce gender biases, could further explain these findings.

These results highlight the importance of continuing to address structural and cultural barriers to ensure sustained equity in sports participation. Future research should explore whether these findings generalize to less structured settings or regions with stronger gender stereotypes.

Additionally, this study identified significant age-related differences in the relationship between passion and basketball engagement, particularly between junior and senior high school students. Harmonious passion positively predicted engagement for junior high students, but this relationship was non-significant for senior high students. Similarly, obsessive passion showed a positive association with engagement among junior high students but not for senior high students.

The decline in passion-engagement relationships among senior high students likely reflects the intense academic pressures associated with China's college entrance exams (Gaokao) [63]. As students transition to senior high, academic demands and parental pressure may overshadow extracurricular activities, reducing the time and energy available for sports participation [64]. This aligns with prior research highlighting how structural academic pressures can diminish adolescents' engagement in physical activities [4,65].

The findings underscore the need for policy interventions that balance academic and physical activity demands during critical educational transitions. Schools could integrate flexible sports programs into senior high curricula to mitigate disengagement. Future studies should examine whether these patterns persist in regions with less stringent academic expectations.

## Practical suggestions

Our findings highlight the need for targeted interventions to enhance basketball participation among Chinese adolescents, particularly for female and senior high school students. Based on the results, we recommend:

(1)  Implementing school-based programs that foster harmonious passion by emphasizing autonomy, enjoyment, and personal growth rather than performance pressure alone. For example, schools can design basketball curricula that offer students choices and set up student-led basketball clubs where students organize training sessions and matches by themselves.

(2) PE teacher education programs should focus on cultivating positive affect and intrinsic motivation through autonomy-supportive strategies, while policy changes should ensure protected time for physical activity across all grade levels. For example, PE teachers can adopt "choice-based task design" in basketball classes, which aims to foster students' motivation in basketball. This strategy directly translate autonomy-supportive principles into teachers' daily instructional practices, which fit the structured context of Chinese school physical education [11].

(3) Basketball coaches should implement choice-based training sessions. This approach aligns with SDT principles by fostering ownership while maintaining technical development goals after school. For example, student players might choose between shooting drills, defensive scenarios, or game-situation simulations during designated training segments.

## Limitations and shortcomings

The current research is limited by sample characteristics, methodological constraints, and cultural considerations.

(1)  The study used convenience sampling and only recruited participants from 1 junior high school and 1 senior high school in southern China, which may limit the generalizability of findings to other regions with different cultural or educational contexts. Future research should employ stratified sampling across diverse geographical and socioeconomic contexts to enhance external validity and better represent China's adolescent population.

(2) The sample has a gender imbalance, with 112 male and 286 female participants. This imbalance may slightly compromise the statistical power of gender-based moderation analyses. Future research is suggested to use stratified

sampling to adjust the gender ratio, which will improve the statistical power of gender moderation analyses and ensure that results are more applicable to both male and female adolescents.

(3) Our reliance on self-report measures, though using validated scales, introduces potential response biases such as social desirability or recall inaccuracy. The cross-sectional design prevents establishing causal relationships between passion, affect, and basketball engagement. A longitudinal design tracking students across multiple years would better capture developmental trajectories and causal pathways.

(4) In the multi – group analysis exploring the moderating effects of gender and age, the model fit was not ideal, with CFI/TLI around.840. This suggests the model might not fully capture the complex relationships, so results related to these moderating effects should be interpreted cautiously. Future research could refine the model or use larger samples to improve fit.

(5) The study focused narrowly on SDT's passion constructs while excluding other potentially relevant factors like peer influence or family support. The Chinese cultural context, with its unique educational pressures and gender norms, may shape passion development differently than Western contexts where much SDT research originates. Future studies should examine how traditional Chinese values interact with motivational processes and explore whether our findings apply to other collectivist cultures.

## Conclusion

This study highlights SDT's utility in understanding youth basketball participation, with harmonious passion and positive affect identified as key drivers of engagement. Physical education curricula and after-school programs should incorporate autonomy-supportive designs that foster intrinsic motivation and harmonious passion while accommodating academic demands. Future research should explore longitudinal and intervention-based designs.

## Author contributions

**Conceptualization:** Jiangchuan Wang.

**Data curation:** Yujie Mao.

**Formal analysis:** Jiangchuan Wang.

**Methodology:** Yujie Mao, Peng Tang.

**Supervision:** Peng Tang.

**Writing – original draft:** Jiangchuan Wang.

**Writing – review & editing:** Peng Tang.

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
