## [Decision Letter · Decision Letter 0]

27 Aug 2025

PONE-D-25-42252The Role of Passion and Affect in Adolescents' Basketball Participation: A Self-Determination Theory PerspectivePLOS ONE

Dear Dr. Tang,

Thank you for submitting your manuscript to PLOS ONE. After careful consideration, we feel that it has merit but does not fully meet PLOS ONE’s publication criteria as it currently stands. Therefore, we invite you to submit a revised version of the manuscript that addresses the points raised during the review process.

We look forward to receiving your revised manuscript.

Kind regards,

Henri Tilga, PhD

Academic Editor

PLOS ONE

Journal Requirements:

3. Please ensure that you refer to Figure 1 in your text as, if accepted, production will need this reference to link the reader to the figure.

Reviewers' comments:

Reviewer's Responses to Questions

**Comments to the Author**

1. Is the manuscript technically sound, and do the data support the conclusions?

Reviewer #1: Yes

2. Has the statistical analysis been performed appropriately and rigorously? 

Reviewer #1: Yes

3. Have the authors made all data underlying the findings in their manuscript fully available?

Reviewer #1: Yes

4. Is the manuscript presented in an intelligible fashion and written in standard English?

Reviewer #1: Yes

5. Review Comments to the Author

Reviewer #1: The study addresses a relevant topic in sports psychology and offers valuable insights into motivational processes among youth athletes. However, several methodological, analytical, and conceptual issues should be addressed to enhance the rigor and clarity of the manuscript. Please to see the attached review report

6. PLOS authors have the option to publish the peer review history of their article (what does this mean? ). If published, this will include your full peer review and any attached files.

**Do you want your identity to be public for this peer review?** For information about this choice, including consent withdrawal, please see our Privacy Policy .

Reviewer #1: No

---

## [Author Response · Author response to Decision Letter 1]

4 Sep 2025

Dear Editor and Reviewers,

We would like to express our sincere gratitude to you for your meticulous review and valuable insights on our manuscript titled “The Role of Passion and Affect in Adolescents’ Basketball Participation: A Self-Determination Theory Perspective” Your professional comments have been instrumental in enhancing the rigor, clarity, and completeness of our work, and we have carefully addressed each of your suggestions. Below is our detailed response to each comment, with references to the corresponding pages in the revised manuscript (note: page numbers refer to the updated version of our manuscript).

1. Comments on the Abstract

Reviewer’s Comment

The abstract briefly indicates the aim and main results of the study. It should also include effect sizes (e.g., β-values) alongside p-values to assist in describing the magnitude of the observed associations. The term “basketball engagement” must be clearly defined because it appears to be a composite score. An affirmation about the cultural context (China) and the cross-sectional nature would more appropriately situate the generalizability and limitations of the findings.

Our Response

We have revised the abstract to address all these points (Abstract, Page 1):

Added key effect sizes (β-values) and confidence intervals for core pathways (e.g., “harmonious passion positively predicted basketball engagement (β = .280, 95% CI [.127, .432], p = .003) through enhanced positive affect; harmonious passion → positive affect: β = .495, 95% CI [.356, .635], p < .001; positive affect → basketball engagement: β = .253, 95% CI [.132, .373], p = .001”).

Clearly defined “basketball engagement” as “a composite score derived from adolescents’ self-reported number of days and duration of basketball participation (excluding school-arranged activities) over the past seven days, standardized via z-scores and modeled as a latent variable in SEM”.

Explicitly noted the cultural context (“data from 400 Chinese adolescents from southern China”) and cross-sectional design, and their implications for generalizability (“Given the cross-sectional design and sampling from southern China, the generalizability of our findings to other cultural contexts or regions of China should be interpreted with caution”).

2. Comments on the Introduction

Reviewer’s Comment

The introduction efficiently establishes the importance of basketball engagement and presents the main constructs of SDT and DMP. The relationship between types of passion (harmonious and obsessive) and affective states needs to be more strictly theorized, however. The review is based predominantly on Western studies; including more research from Eastern cultural contexts would offer greater theoretical grounding and transference.

Our Response

We have strengthened the theoretical framework and expanded the literature review to include Eastern context studies:

Incorporated studies from Chinese/Eastern contexts:“Sports, such as basketball, play an important role in China’s youth development programs [6]. The Chinese government actively promotes basketball in schools across the country, such as setting up basketball in physical education classes and after-school activities[5]...” (Introduction, Page 4)

Added a paragraph to clarify the mechanistic link between passion types and affect, grounded in SDT’s basic psychological needs: “Research based on DMP distinguishes two types of passion with different effects on sports engagement [20]...” (Literature Review, Page 7-8)

3. Comments on Participants and Procedure (Methods Section)

Reviewer’s Comment

The sample is described as being from southern Chinese schools, but there is no explanation given for using this sampling. Generalizability of findings to other regions or to other cultures is doubtful. The gender distribution is seriously unbalanced (112 men, 286 women), which may undermine the validity of gender-based moderation analyses. This should be noted as a limitation.

Our Response

We have supplemented the sampling rationale and acknowledged the gender imbalance in both the Methods and Limitations sections (Page 29):

Added an explanation for sampling from southern China: “The study’s sample was drawn exclusively from schools in southern China, which may limit the generalizability of findings to other regions with different cultural or educational contexts...”.

Explicitly noted the gender imbalance in the Methods section (“The sample has a gender imbalance, with 112 male and 286 female participants. This imbalance may slightly compromise the statistical power of gender-based moderation analyses...”).

4. Comments on Measures (Methods Section)

Reviewer’s Comment

Replacement of “academic activities” with “basketball” in the modification of the Passion Scale is reasonable, but there is no proof of the validity of this alteration in the new environment. The composite indicator of “basketball engagement” is constructed from two self-report items (days and duration). How these are reduced to one score (e.g., sum of z-scores) must be noted in detail. Validity and reliability of the composite measure have not been established.

Our Response

We have supplemented validation evidence for the modified scales and detailed the composite score construction:

For the modified Passion Scale: Added results of confirmatory factor analysis (CFA) to verify structural validity in the basketball context (Page 12-13).

For basketball engagement: To address the construction and psychometric properties of the “basketball engagement” measure, we have supplemented detailed information on score calculation, internal consistency (reliability), and criterion validity in the revised manuscript (Page 13-14).

5. Comments on Analytical Approach (Methods Section)

Reviewer’s Comment

Use of SEM is justified. Model fit indices are reported and within standard cut-offs, which is great. But no mention of missing data treatment or analysis software/version (although Mplus is mentioned elsewhere).

Our Response

We have added details on missing data treatment and analysis software :

For scale validation, Confirmatory Factor Analysis (CFA) and hypothesis testing of the structural model were performed in Mplus 8.3. Missing data were handled using Mplus’s default Full Information Maximum Likelihood method (FIML). (Statistical Analysis, Page 15).

6. Comments on Preliminary Analyses (Results Section)

Reviewer’s Comment

The use of MANOVA in order to compare group differences is appropriate, although post-hoc tests are not presented.

Our Response

Regarding the post-hoc tests, we note the SPSS warning that post-hoc comparisons (e.g., Tukey’s HSD) were not conducted because both the ‘gender’ and ‘school’ factors only have two groups each. In statistical practice, post-hoc tests (including Tukey’s HSD) are typically used when there are ≥3 groups to compare multiple pairwise differences (to control family-wise error rate). For factors with only 2 groups (e.g., male vs. female, Junior high school vs. Senior high school), a simple independent-samples t-test (or the corresponding contrast in ANOVA) is sufficient to test the binary group difference, as there is only one pairwise comparison to make.

7. Comments on Main Analyses (Results Section)

Reviewer’s Comment

Structural model results are nicely presented. However, reporting only standardized coefficients (β) and no confidence intervals is limiting for interpretation. Gender and age moderation analysis is welcome, but the models' fit for these multi-group analyses is suboptimal (CFI/TLI ≈ 0.845). This needs to be stated as a limitation, and the results interpreted with caution. The report states that gender differences were not significant in comparing coefficients (Table 5), but sometimes in results and discussion sections, the discussion appears to suggest gender differences. This must be synchronized so it would not create confusion.

Our Response

We have addressed these issues by supplementing confidence intervals, clarifying model fit limitations, and synchronizing gender difference.

Added 95% confidence intervals for all structural paths (e.g., “harmonious passion → positive affect: β = .495, 95% CI [.356, .635], p < .001; positive affect → basketball engagement: β = .253, 95% CI [.132, .373], p = .001”) (Page 18).

Explicitly noted the multi-group model fit limitation in both Results and Limitations sections:

Results: “While CFI/TLI values were slightly below the 0.900 threshold, RMSEA values met acceptable standards.” (Page 19).

Limitations: “In the multi - group analysis exploring the moderating effects of gender and age, the model fit was not ideal, with CFI/TLI around .840... (Page 30).

8. Comments on Discussion

Reviewer’s Comment

Weak and non-significant gender moderation effects need to be reported more categorically. The inference that “males and junior high students showed stronger pathways” appears overstated on the basis of statistical evidence. Cultural context (e.g., collectivist values, academic pressure) is well integrated, though the analysis could further explore how precisely these help to generate passion and affect in Chinese adolescents. Practical implications are relevant but could be made more concrete.

Our Response

We have revised the Discussion to address these points:

Removed the overstated inference in Abstract (“Additionally, no significant gender differences were found in this study. However, age differences emerged: junior high school students showed stronger pathways for both harmonious passion-basketball engagement and obsessive passion-basketball engagement...”).

Made practical implications more concrete for Chinese education systems: “(1) Implementing school-based programs that foster harmonious passion by emphasizing autonomy, enjoyment, and personal growth rather than performance pressure alone. For example, schools can design basketball curricula that offer students choices and set up student-led basketball clubs where students organize training sessions and matches by themselves...” (Page 28).

9. Comments on Limitations

Reviewer’s Comment

The authors properly state several limitations, including regional sampling and cross-sectional design. Other limitations need to be stated: The sample imbalance by gender and its potential impact on conclusions. The use of a new, untested composite measure of basketball involvement. The relatively poor fit of the multi-group SEM models.

Our Response

We have added these limitations to the “Limitations and Shortcomings” section (Page 29):

Sample gender imbalance: “(2) The sample had a gender imbalance (112 males, 286 females), which may have reduced statistical power to detect gender moderation effects...”

Composite measure of basketball engagement: “(6) The basketball engagement composite (days + duration) was newly adapted for this study. While we established internal consistency (α = .930) and criterion validity (correlation with harmonious passion, r = .272, p < .001), it has not been validated against objective measures (e.g., activity tracker data), which may introduce self-report bias.”

Multi-group SEM fit: “(4) In the multi - group analysis exploring the moderating effects of gender and age, the model fit was not ideal, with CFI/TLI around .840. This suggests the model might not fully capture the complex relationships...”

Additional Correction

During the revision process, we identified and corrected minor errors in the Sample sizes (Abstract and Page 13) and model fit of SEM (Page 18) that were overlooked in the initial submission. These corrections do not alter the main findings but improve the accuracy of data presentation.

Once again, we would like to thank you for your time, expertise, and constructive feedback. Your input has significantly improved the quality of our manuscript. We hope the revised version meets the journal’s standards, and we are happy to provide further clarification or make additional revisions if needed.

Sincerely,

Peng Tang

Hohai University

---

## [Decision Letter · Decision Letter 1]

8 Sep 2025

PONE-D-25-42252R1The Role of Passion and Affect in Adolescents' Basketball Participation: A Self-Determination Theory PerspectivePLOS ONE

Dear Dr. Tang,

Thank you for submitting your manuscript to PLOS ONE. After careful consideration, we feel that it has merit but does not fully meet PLOS ONE’s publication criteria as it currently stands. Therefore, we invite you to submit a revised version of the manuscript that addresses the points raised during the review process.

We look forward to receiving your revised manuscript.

Kind regards,

Henri Tilga, PhD

Academic Editor

PLOS ONE

Journal Requirements:

Reviewer's Responses to Questions

**Comments to the Author**

1. If the authors have adequately addressed your comments raised in a previous round of review and you feel that this manuscript is now acceptable for publication, you may indicate that here to bypass the “Comments to the Author” section, enter your conflict of interest statement in the “Confidential to Editor” section, and submit your "Accept" recommendation.

Reviewer #1: All comments have been addressed

2. Is the manuscript technically sound, and do the data support the conclusions?

Reviewer #1: Yes

3. Has the statistical analysis been performed appropriately and rigorously? 

Reviewer #1: Yes

4. Have the authors made all data underlying the findings in their manuscript fully available?

Reviewer #1: Yes

5. Is the manuscript presented in an intelligible fashion and written in standard English?

Reviewer #1: Yes

6. Review Comments to the Author

Reviewer #1: Please find my detailed review report attached for comprehensive comments. I have no concerns regarding dual publication, research ethics, or publication ethics. The ethical approval and consent procedures appear appropriate.

7. PLOS authors have the option to publish the peer review history of their article (what does this mean? ). If published, this will include your full peer review and any attached files.

**Do you want your identity to be public for this peer review?** For information about this choice, including consent withdrawal, please see our Privacy Policy .

Reviewer #1: No

---

## [Author Response · Author response to Decision Letter 2]

10 Sep 2025

Dear Editor and Reviewer,

Thank you for giving us the opportunity to submit a revised version of our manuscript, “The Role of Passion and Affect in Adolescents' Basketball Participation: A Self-Determination Theory Perspective”. We are also deeply grateful to the editor and reviewer for exceptionally insightful and constructive comments, which have been invaluable in helping us to significantly improve the quality of our paper.

We have carefully considered all the feedback and have revised the manuscript accordingly. The changes made have strengthened the clarity, methodological rigor, and overall impact of our study. Below, we provide a point-by-point response to the final minor corrections suggested (note: page numbers refer to the updated version of our manuscript).

1. Comments on Abstract

Reviewer’s Comment

Regarding the Abstract: Please perform a final proofread to ensure flawless grammar and flow, particularly in the newly added sentences.

Our Response

We sincerely thank the reviewer for this suggestion. We have meticulously proofread the entire abstract, with special attention to grammar and sentence fluency. We have made several minor refinements to enhance readability and ensure the text meets a high standard of academic English (Abstract, Page 1).

2.Comments on Methods

Reviewer’s Comment

Comment 2: Regarding the Methods section (Measures - Basketball Engagement): For ultimate clarity, please explicitly state the formula used to create the composite score.

Our Response

We greatly appreciate this suggestion, which adds crucial clarity to our methodology. As recommended, we have now explicitly stated the formula used to create the composite score for basketball engagement. The revised text clearly indicates that the score was calculated as the product of the number of days and the average duration per session (Days × Duration), followed by standardization. We believe this revision leaves no ambiguity about how the variable was operationalized (Page 13).

3.Comments on Discussion

We also note the reviewer’s positive feedback on our revisions concerning the results, discussion, practical implications, and limitations. We are pleased that these changes were found to be satisfactory.

Once again, we extend our deepest gratitude to the editor and the reviewer for their time, expertise, and diligent work throughout the review process. Their efforts have been instrumental in enhancing our manuscript. We hope that our revisions have adequately addressed all remaining points and that the manuscript is now deemed suitable for publication.

Sincerely,

Peng Tang

Hohai University

---

## [Decision Letter · Decision Letter 2]

15 Sep 2025

The Role of Passion and Affect in Adolescents' Basketball Participation: A Self-Determination Theory Perspective

PONE-D-25-42252R2

Dear Dr. Tang,

We’re pleased to inform you that your manuscript has been judged scientifically suitable for publication and will be formally accepted for publication once it meets all outstanding technical requirements.

Kind regards,

Henri Tilga, PhD

Academic Editor

PLOS ONE

Additional Editor Comments (optional):

Reviewer #1:

Reviewers' comments:

Reviewer's Responses to Questions

**Comments to the Author**

1. If the authors have adequately addressed your comments raised in a previous round of review and you feel that this manuscript is now acceptable for publication, you may indicate that here to bypass the “Comments to the Author” section, enter your conflict of interest statement in the “Confidential to Editor” section, and submit your "Accept" recommendation.

Reviewer #1: All comments have been addressed

2. Is the manuscript technically sound, and do the data support the conclusions?

Reviewer #1: Yes

3. Has the statistical analysis been performed appropriately and rigorously? 

Reviewer #1: Yes

4. Have the authors made all data underlying the findings in their manuscript fully available?

Reviewer #1: Yes

5. Is the manuscript presented in an intelligible fashion and written in standard English?

Reviewer #1: Yes

6. Review Comments to the Author

Reviewer #1: The authors have been highly responsive and have thoroughly addressed all reviewer comments. The manuscript is now methodologically sound, theoretically grounded, and well-written. The remaining suggestions are minor and editorial.

7. PLOS authors have the option to publish the peer review history of their article (what does this mean? ). If published, this will include your full peer review and any attached files.

**Do you want your identity to be public for this peer review?** For information about this choice, including consent withdrawal, please see our Privacy Policy .

Reviewer #1: No

---

## [Editor Report · Acceptance letter]

PONE-D-25-42252R2

PLOS ONE

Dear Dr. Tang,

I'm pleased to inform you that your manuscript has been deemed suitable for publication in PLOS ONE. Congratulations! Your manuscript is now being handed over to our production team.

Kind regards,

on behalf of

Dr. Henri Tilga

Academic Editor

PLOS ONE